# Hausdorff Fractal Derivative Model to Characterize Transport of Inorganic Arsenic in Porous Media

**Xiaoxiao Hao [1], HongGuang Sun [1,\*], Yong Zhang [2], Shiyin Li [3], Jia Song [3] and Kate Salsky [2]**

[1] State Key Laboratory of Hydrology-Water Resources and Hydraulic Engineering, College of Mechanics and Materials, Hohai University, Nanjing 210098, China; xxhao@hhu.edu.cn

[2] Department of Geological Sciences, University of Alabama, Tuscaloosa, AL 35487, USA; yzhang264@ua.edu (Y.Z.); kesalsky@crimson.ua.edu (K.S.)

[3] School of Environment, Nanjing Normal University, Nanjing 210023, China; lishiyin@njnu.edu.cn (S.L.); sj13276685877@163.com (J.S.)

\* Correspondence: shg@hhu.edu.cn; Tel.: +86-136-2158-6259

**Abstract:** The increasing severity of arsenic pollution has progressively threatened human life and attracted much attention. One of the important topics in environmental sciences is to accurately describe the inorganic arsenic transport in heterogeneous porous media, occurring anomalous diffusion phenomenon, which ultimately benefits the control of arsenic pollution. In this paper, we re-evaluate the dataset of the inorganic arsenic transport in porous media in previous work by using a time-Hausdorff fractal model (HADE). Transport experiments of arsenic-carrying (As(V)) ferric humate complex colloids through a quartz sand column were carried out under varying dissolved organic matter (humic acid) concentrations, pH values, ionic strengths, and ferric concentrations. The results show that under our experimental settings, arsenic migration is promoted with the increase of concentrations of HA, ferric ion and sodium ion, and pH to varying degrees. The intensity of arsenic sub-diffusion behavior is opposite to that of arsenic transport. The HADE model can describe the migration behavior of arsenic well, and the value of the time fractal derivative can reflect the diffusion intensity of arsenic migration to a certain extent. By comparing the HADE model, ADE model, and time-fractional model (fADE) to the experimental data, the HADE model can significantly improve all the simulation results of capturing As(V) breakthrough curves (BTCs).

**Keywords:** Hausdorff fractal derivative model; arsenic transport; heterogeneity

## 1. Introduction

Heavy metal pollution in soil threatens human health and the environment, which is a global issue drawing broad attention [1,2]. Among the heavy metal pollutants, Arsenic(As) is a natural semi-metallic element that occurs in the earth's crust, carried in the sulfur-arsenic ore, or associated with Cu, Pb, and/or Zinc sulfides [3,4]. In addition to the naturally occurring and highly toxic As, a large amount of arsenic has been abundant in the environment after industrial processes such as glass and paper production, coal mining, and the semiconductor industry [5,6]. After a series of transformations and migrations, arsenic can become major contamination in the environment [7]. For pollution control and soil remediation purposes, it is necessary to fully understand the fate and transport of As(V) in the environment, particularly the reliable quantification of As(V) dynamics in saturated soil on which this study is mainly focused.

Many efforts have been devoted to exploring the migration of arsenate in soils [8–10]. The influences of both, humic acid and iron oxide colloids, on the transport of arsenic in soil, are significantly important. Natural organic matter (NOM) and iron minerals are widely abundant in

groundwater and soil system in the form of soil colloids. In the natural environment, arsenic is easily combined with colloids as it forms a ternary complex with natural organic matter and iron oxide minerals [11,12]. Humic acid (HA) is the main component of soil organic matter, comprising numerous functional groups such as hydroxyl groups and carboxyl groups, and plays an important role in regulating the migration and transformation of pollutants in the environment. Luo et al. found that humic acid (HA) can be strongly adsorbed into the exchange surfaces of iron oxides to form HA-iron oxide, and therefore can compete with As(V), which affects the adsorption of As(V) into the exchange sites of the iron oxides [13]. Besides, the arsenic adsorption is greatly influenced by pH, having strong retention in lower pH value (less than 7), and then affects the transport behavior [14]. Sharma et al. [15] showed that the presence of organic matter and iron oxide minerals enhance the migration of arsenic in their conducted column experiments. The effect of colloids (including the soil colloid and ferrihydrite colloid) on the migration of arsenic in aqueous media was investigated by Ma et al. [16] under different environmental factors, revealing that soil colloids were resistant to arsenic adsorption under both, neutral and alkaline conditions.

As an attempt to better describe the migration of arsenic through natural systems, multiple models have been applied. Zhang and Selim [17] used the multi reaction transport model (MRM) to describe the asymmetrical and retarded BTCs for As(V) by conducting arsenate sorption and transport column experiments in three different soils. Lim et al. [18] established a mathematical model for characterizing the transport of arsenic combined with biogeochemical processes. Their results indicated that the model was more compatible with the experimental data due to the inclusive term describing the precipitation process. Huang et al. [19] used a one-dimensional advection-dispersion equation (ADE) model to compare and analyze the migration behavior of As(III) and As(V) in bone charcoal, and they obtained great modeling results. Similarly, Giménez et al. [20] also simulated the migration and adsorption process of arsenic on natural hematite by using the CXTFIT code based on the ADE model. To simulate the leaching ability of arsenic in lake sediments close to a landfill in Maine, a kinetic model was used by Nikolaos et al. [21], to describe the geochemistry and the rate of the mobility of the arsenic. Besides, a numerical model was developed by Eljamal et al., adding chemical reaction into 1-D solute transport, to simulate the permeable reactive barrier column results, including the oxidation and sorption of arsenic [22].

As a tool for characterizing the complexity of porous media, the fractal theory has been recently applied in various fields [23–26]. Heymans et al. [27] applied the fractal rheological model to describe the viscoelastic behavior under various stress modes, and proved the equivalence between tree and ladder fractal models; Liang et al. [28] introduced the Fick's law for diffusion using fractal time and space derivatives and linked the tissue components of subsomin to the attenuation of MRI signals observed after the application of diffusion sensitive pulse sequences. Ryutaro Kanno [29] proposed a statistical representation in arbitrary fractal space-time, analyzing the anomalous diffusion of fractal structures, and analyzed random walks on 2D Sierpinski which led to great results. Porous media, turbulence, and other media usually exhibit fractal characteristics. However, classical physical laws based on Euclidean geometry such as Fick's Law, cannot accurately describe physical processes in porous media with fractal dimension. Therefore, it is necessary to redefine basic physical concepts, such as distance and velocity in fractal media. The space-time scale should be transformed according to $(x^\beta, t^\alpha)$. The fractal models appear to have great adaptability in modeling the anomalous diffusion in porous media like water transport. To better describe diffusion and transport behavior in porous media, Chen et al. [30] proposed the Hausdorff fractal derivative to associate the anomalous transport model with the derived fundamental solution. Based on the redefinition of the physical concept in fractal media, Chen et al. defined the Hausdorff fractal derivative for time and space (Equations (1) and (2)). They conducted a comparison between the Hausdorff fractal derivative model and the fractional derivative model in terms of the calculation efficiency, diffusion rate, and heavy-tail characteristics to validate its efficiency [31]. The results showed that the Hausdorff fractal derivative model is simple and more efficient than the standard fractional derivative model in characterizing

the anomalous diffusion. Sun et al. [32] further employed the Hausdorff fractal derivative to build a fractal Richards' equation, providing a better fit for water content curves in unsaturated zones than the classical Richards' equation.

As the fractal model can better describe the anomalous diffusion behavior, also considering the time dependence of the diffusion, this study aims to apply the time-Hausdorff fractal model (HADE) based on classical ADE model to (1) characterize inorganic arsenic transport in porous media and investigate the influences of the factors on the anomalous diffusion of arsenic migration; (2) discover their connection with time fractal derivative; (3) compare with ADE model and time-fractional model to discuss its applicability and advantages, to eventually provide a new descriptive framework for the transport of arsenic-containing pollutants. Previous laboratory experiments were conducted to monitor the dynamics of As(V) moving in saturated soils under four selected variables, and the results were analyzed for the phenomenon, which were simply modeled by using fractional model [33]. This paper was focusing on the new model (HADE), the comparison of the HADE model with other models (ADE, fADE), and the application of experimental data to verify the effectiveness and advantages of the new model.

## 2. Experimental Method

In this experiment, As(V) and ferric humate complex was designated as the pollutant and the typical dissolved organic matter (DOM) respectively, while quartz sand was used to build the porous media system. Previous studies showed that humic acid, pH, ferric iron concentration, and iron salt concentration are the four main possible factors affecting the transport of As in DOM [33,34], and therefore were considered in the experiment. In this section, we will only give the necessary experimental information. For more details, please check our previous work [33].

### 2.1. Preparation of the Experiment

The background solutions were: 500 mg/L HA solution, 1000 mg/L sodium arsenate solution, the ferric nitrate colloidal stock solution containing 1 mmol/L of ferric ions, and 0.01 mol/L $NaNO_3$ solution.

After pre-treating the 40–60 mesh quartz sand by removing impurities, washing the sand with ionized water and drying it, small quantities of quartz were taken many times to fill the column with quartz sand until the peristaltic column, with a diameter of 2.6 cm and a length of 20.0 cm, was filled.

### 2.2. Arsenate Transport Experiments

The experimental device diagram is shown in Figure 1. The variables of 4 runs of the experiments and the other three controlled influencing factors are shown in Table 1. The following is a detailed description of each run of experiments.

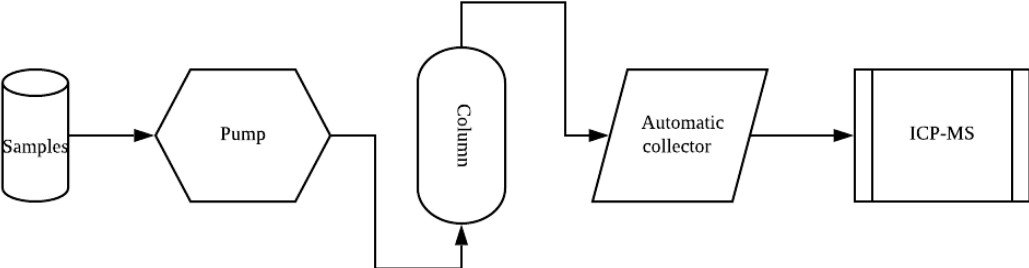

**Figure 1.** Experimental setup.

**Table 1.** Experimental variables in the experiments.

| Variables | Run 1 | Run 2 | Run 3 | Run 4 |
|---|---|---|---|---|
| Humic Acid (mg/L) | 0, 5, 10 | 5 | 5 | 5 |
| pH | 6.0 | 4.5, 6.0, 7.5 | 6.0 | 6.0 |
| Ironic Strength (mol/L) | 0.01 | 0.01 | 0.001, 0.01, 0.1 | 0.01 |
| Ferric nitrate (mmol/L) | 0.025 | 0.025 | 0.025 | 0, 0.0125, 0.025 |

Run 1: Effect of humic acid on the migration of As(V) carried by DOM-Fe ($Fe^{3+}$) composite colloids

A mixture of 0.025 mmol/L ferric nitrate and HA of different concentrations (0, 5, and 10 mg/L) was prepared. Sodium arsenate was added to set the concentration of As(V) in the mixed solution up to 1 mg/L. The pH value was adjusted to 6.0 using $NaNO_3$ (with the ionic strength of 0.01 mol/L) as the supporting electrolyte.

Run 2: Effect of pH on the migration of As(V) carried by DOM-Fe ($Fe^{3+}$) composite colloids

The procedures are similar to those in Run 1, except that the pH value was adjusted to 4.5, 6.0 and 7.5, respectively, with nitric acid or sodium hydroxide, while using $NaNO_3$ (with the ionic strength of 0.01 mol/L) as the supporting electrolyte.

Run 3: Effect of sodium the migration of As(V) carried by DOM-Fe ($Fe^{3+}$) composite colloids

The procedures are similar to those in Run 1, except that $NaNO_3$ with various ionic strengths (0.001, 0.01, and 0.1 mol/L respectively) were used as supporting electrolytes, and the pH value was adjusted to 6.0.

Run 4: Effect of ferric on the migration of As(V) carried by DOM-Fe ($Fe^{3+}$) composite colloids

A mixture of HA (with a concentration of 5 mg/L) and ferric nitrate (with the concentration increasing from 0, 0.0125, to 0.025 mmol/L) were prepared. The other steps are similar to those in Run 1.

The measured breakthrough curve (BTC) for As(V) during each experimental run is illustrated by symbols in Figures 2–5.

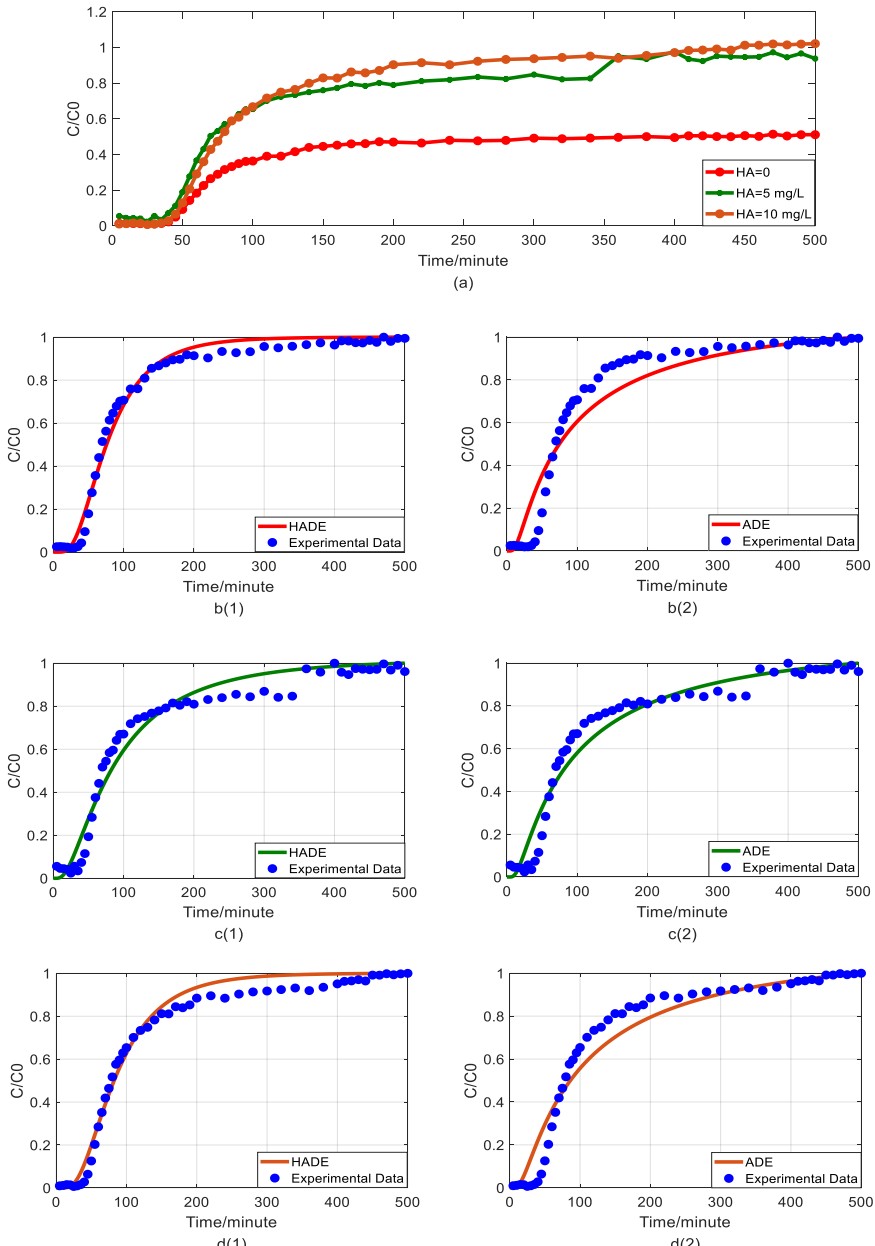

**Figure 2.** Impact of humic acid (HA) concentration on As(V) transport: (**a**) Breakthrough curves at three HA concentrations; (**b1,b2**): time-Hausdorff fractal model (HADE) and advection-dispersion equation (ADE) fitting results when the HA concentration is 0; (**c1,c2**): HADE and ADE fitting results when the concentration is 5 mg/L; (**d1,d2**): HADE and ADE fitting results when the concentration is 10 mg/L.

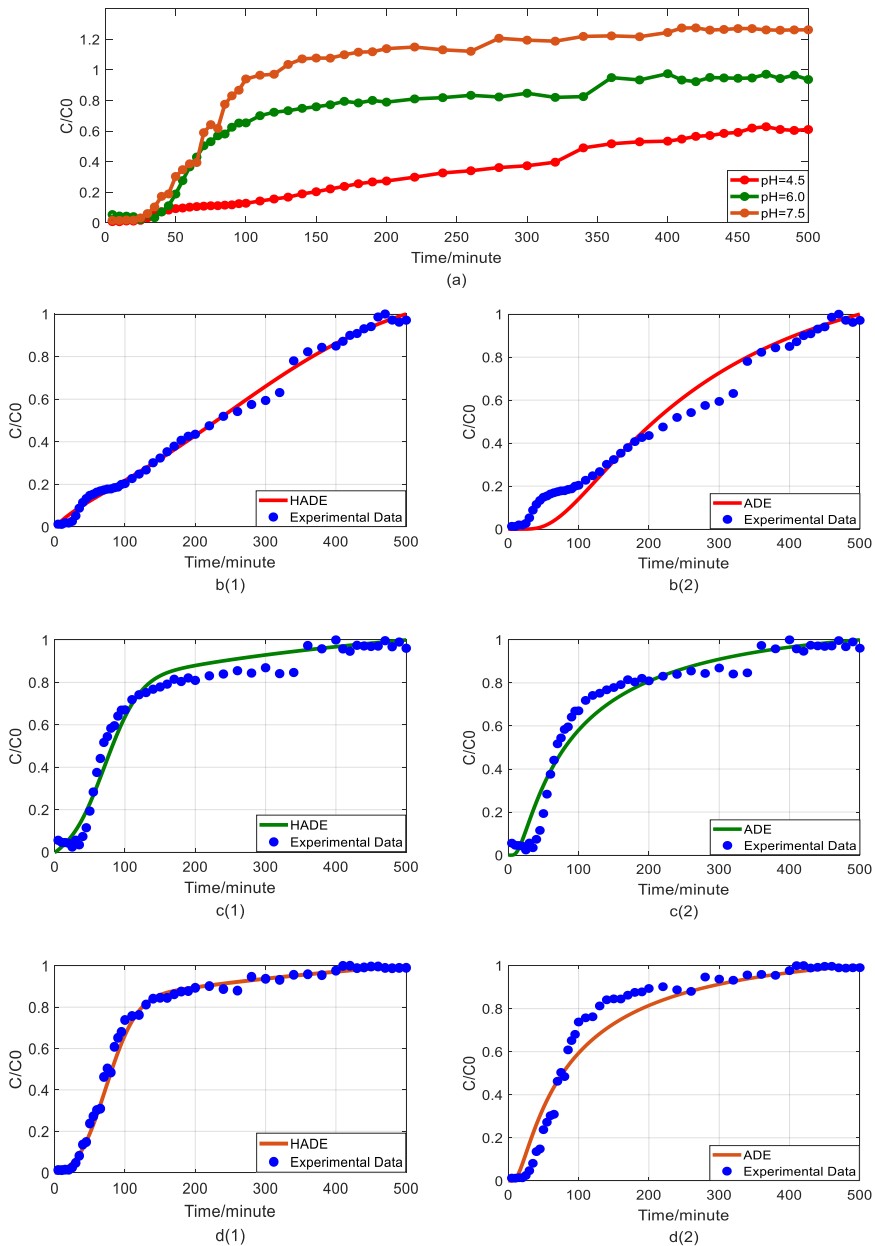

**Figure 3.** Impact of pH values on As(V) transport: (**a**) breakthrough curves at three pH values; (**b1,b2**): HADE and ADE fitting results when the pH is 4.5; (**c1,c2**): HADE and ADE fitting results when the pH 6.0; (**d1,d2**): HADE and ADE fitting results when the pH is 7.5.

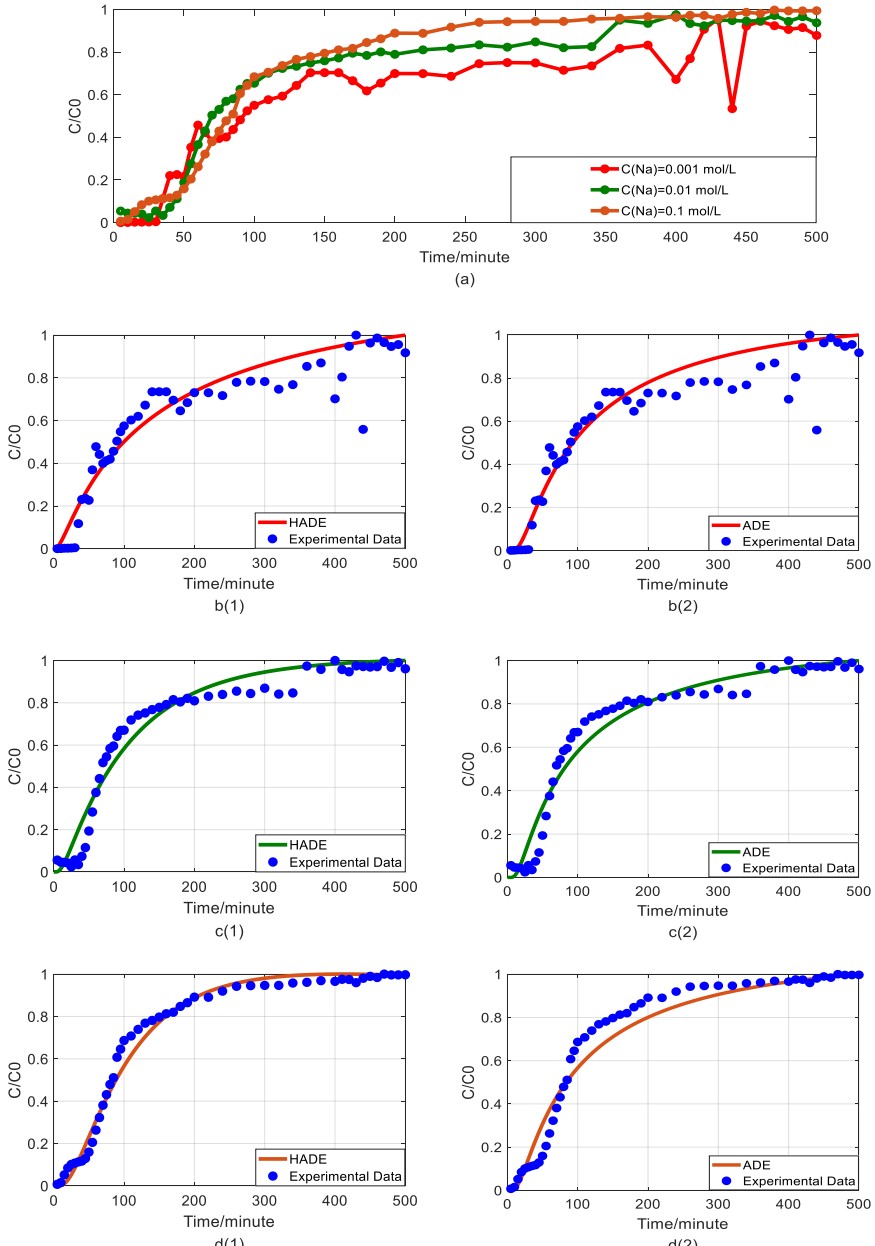

**Figure 4.** Impact of sodium concentrations on As(V) transport: (**a**) Breakthrough curves at three concentrations of sodium; (**b1,b2**): HADE and ADE fitting results when the concentration of sodium is 0.001 mol/L; (**c1,c2**): HADE and ADE fitting results when the concentration of sodium is 0.01 mol/L; (**d1,d2**): HADE and ADE fitting results when the concentration of sodium is 0.1 mol/L.

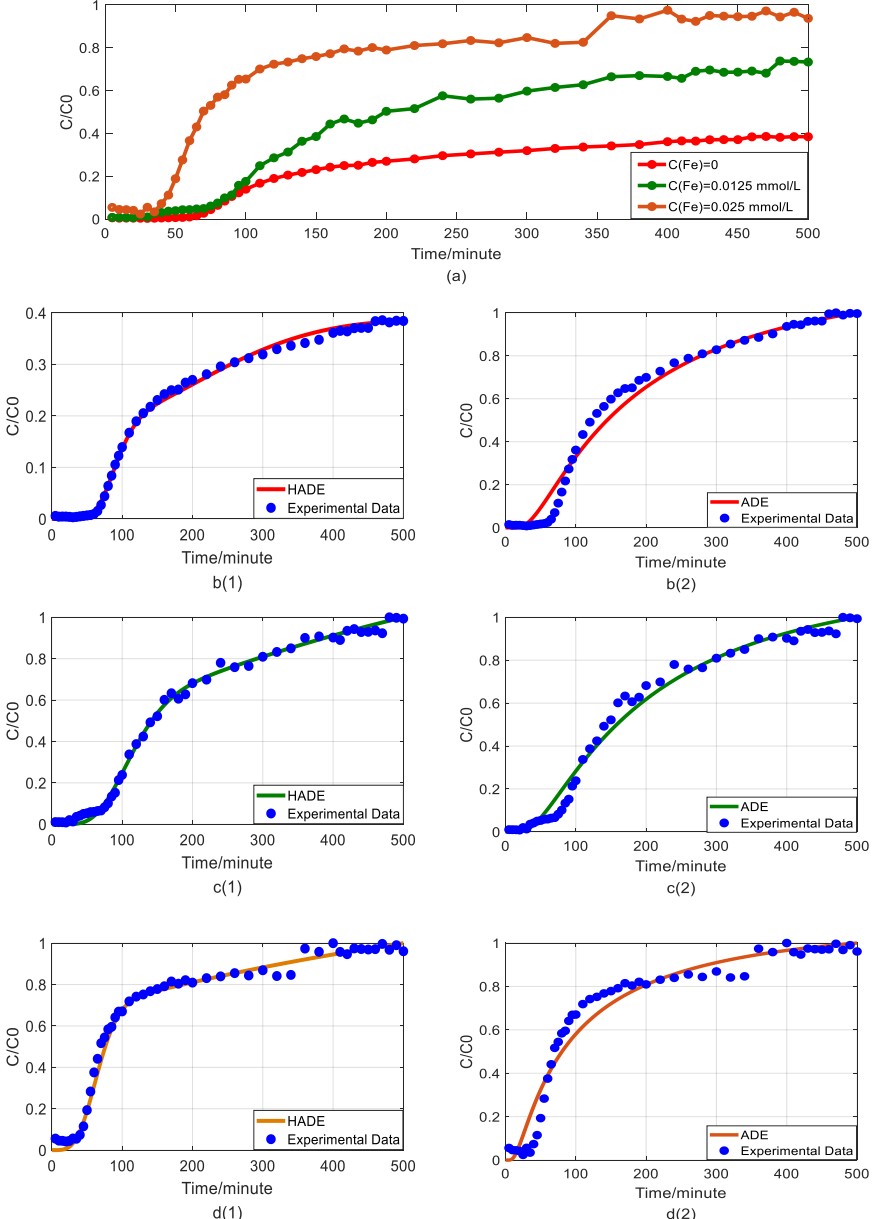

**Figure 5.** Impact of iron concentrations on As(V) transport: (**a**) Breakthrough curves at three concentrations of iron; (**b1,b2**): HADE and ADE fitting results when the concentration of iron is 0; (**c1,c2**): HADE and ADE fitting results when the concentration of iron is 0.0125 mmol/L; (**d1,d2**): HADE and ADE fitting results when the concentration of iron is 0.025 mmol/L.

## 3. Hausdorff Fractal Derivative Model

As reviewed above, the Hausdorff fractal derivative has been applied to solve dynamics observed in hydraulics, creep, anomalous diffusion, petroleum, and economics [27]. The definition of Hausdorff fractal derivative is provided based on the Hausdorff fractal space-time distance in one-dimension defined below:

$$\begin{cases} \Delta \hat{t} = \Delta t^{\alpha}, \\ \Delta \hat{x} = \Delta x^{\beta}, \end{cases} \tag{1}$$

where $\alpha$ is the time fractal index, and $\beta$ is the space fractal index, t [T] and x [L] represent the time and space respectively. The Hausdorff fractal derivative is then defined as

$$
\begin{cases}
\frac{du}{dx^{\beta}} = \lim\limits_{x\prime \to x} \frac{u(x)-u(x\prime)}{x^{\beta}-x\prime^{\beta}}, \\
\frac{du}{dt^{\alpha}} = \lim\limits_{t\prime \to t} \frac{u(t)-u(t\prime)}{t^{\alpha}-t\prime^{\alpha}},
\end{cases}
\tag{2}
$$

The classical ADE model has been long used by hydrologists to study solute transport in porous media. The one-dimensional ADE model can be expressed as

$$
\frac{\partial C}{\partial t} + u\frac{\partial C}{\partial x} = K\frac{\partial}{\partial x}\left(\frac{D}{K}\frac{\partial C}{\partial x}\right),
\tag{3}
$$

where C is a continuous function defining the solute concentration, u $[LT^{-1}]$ is the average fluid velocity, K $[L\,T^{-1}]$ is the hydraulic conductivity, and D $[L^2\,T^{-1}]$ is the dispersion coefficient.

However, the ADE model cannot accurately describe solute transport in most porous media, due to the multi-scale heterogeneity of natural media [35–38]. The following time Hausdorff fractal derivative model, which fully considers the fractal property or heterogeneity of the considered media, may serve as an alternative modeling approach:

$$
\frac{\partial C}{\partial t^{\alpha}} + u\frac{\partial C}{\partial x} = K\frac{\partial}{\partial x}\left(\frac{D}{K}\frac{\partial C}{\partial x}\right),
\tag{4}
$$

Analytic solution of the Hausdorff fractal ADE model (HADE) can be obtained using a similar method for the classical ADE model with the time variable substituted for $t^{\alpha}$ [34]:

$$
\frac{C}{C_0} = \frac{1}{2}\left[\operatorname{erfc}\left(\frac{x-ut^{\alpha}}{(4Dt^{\alpha})^{\frac{1}{2}}}\right) + \exp\left(\frac{xu}{D}\right)\bullet\operatorname{erfc}\left(\frac{x+ut^{\alpha}}{(4Dt^{\alpha})^{\frac{1}{2}}}\right)\right],
\tag{5}
$$

where K in the Equation (4) is equal to 1, D and u are constants in the region of [0, ∞], erfc(x) is the complementary error function, which is defined as:

$$
\operatorname{erfc}(x) = \frac{1}{\sqrt{\pi}}\int_{-x}^{x} e^{-t^2}dt,
\tag{6}
$$

It should be mentioned that due to the redefinition of the fractal dimension of time, the D and u in HADE have also been newly modified on the time fractal, which is different from the D and u in classical ADE.

And the boundary conditions are

$$
\begin{aligned}
&C(0,t) = C_0 = \text{Constant } t > 0, \\
&\lim_{x\to\infty} C(x,t) = 0 \text{ t} > 0
\end{aligned}
\tag{7}
$$

And the initial condition is

$$
C(x,0) = 0 \quad 0 \le x \le \infty
\tag{8}
$$

## 4. Results

We tested the applicability of the temporal HADE model using the experimental data obtained and investigated its superiority compared with the classical ADE model. The results are depicted in Figures 2–5. For the convenience of the readers, we still gave the original results of the experiments (Figures 2a, 3a, 4a and 5a). Besides, we simply discussed the comparison between the HADE model and the fADE model [33] to convince the advantages of our new model, and the results are shown in Figure 6. To better reflect the fitting results and quantitatively compare the advantages of the two

models, we normalized the experimental results and calculated the global error (GE). The function of the GE is

$$\mathrm{GE} = \left[\sum_{k=1}^{M} \left[I_{\mathrm{numer}}(k) - I_{\mathrm{exact}}(k)\right]^2\right]^{\frac{1}{2}} / \left[\sum_{k=1}^{M} \left[I_{\mathrm{exact}}(k)\right]^2\right]^{\frac{1}{2}} \tag{9}$$

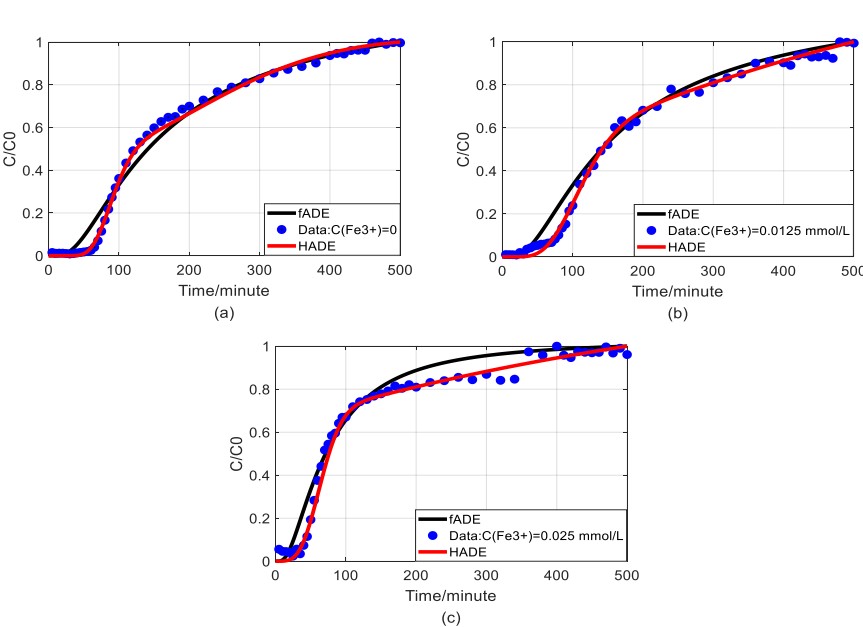

**Figure 6.** Comparison of fADE and HADE models of As(V) transport with the impact of iron concentrations (flow rate is 2 mL/min): (**a**) HADE and fADE fitting results when the concentration of iron is 0; (**b**) HADE and fADE fitting results when the concentration of iron is 0.0125 mmol/L; (**c**) HADE and fADE fitting results when the concentration of iron is 0.025 mmol/L.

The model's simulation parameters are shown in Tables 2–6.

**Table 2.** Fitting parameters and errors at three HA concentrations.

| C (HA) | 0 | 5 mg/L | 10 mg/L |
|---|---|---|---|
| $\alpha$ (HADE) | 0.4 | 0.48 | 0.49 |
| D (HADE) | 0.04 | 0.08 | 0.04 |
| GE (HADE) | 0.054167 | 0.087855 | 0.062635 |
| D (ADE) | 0.057 | 0.051 | 0.046 |
| GE (ADE) | 0.12359 | 0.098227 | 0.1083 |

**Table 3.** Fitting parameters and errors at three pH values.

| pH | 4.5 | 6.0 | 7.5 |
|---|---|---|---|
| $\alpha$ (HADE) | 0.13 | 0.48 | 0.49 |
| D (HADE) | 0.15 | 0.08 | 0.05 |
| GE (HADE) | 0.052165 | 0.087855 | 0.052928 |
| D (ADE) | 0.01 | 0.051 | 0.054 |
| GE (ADE) | 0.13004 | 0.098227 | 0.10396 |

**Table 4.** Fitting parameters and errors at three sodium concentrations.

| C (Sodium) | 0.001 mol/L | 0.01 mol/L | 0.1 mol/L |
|---|---|---|---|
| $\alpha$ (HADE) | 0.41 | 0.48 | 0.49 |
| D (HADE) | 0.18 | 0.08 | 0.05 |
| GE (HADE) | 0.14957 | 0.087855 | 0.050204 |
| D (ADE) | 0.041 | 0.051 | 0.048 |
| GE (ADE) | 0.15134 | 0.098227 | 0.092982 |

**Table 5.** Fitting parameters and errors at three iron concentrations.

| C (Iron) | 0 | 0.0125 mmol/L | 0.025 mmol/L |
|---|---|---|---|
| $\alpha$ (HADE) | 0.72 | 0.6 | 0.4 |
| D (HADE) | 0.011 | 0.015 | 0.1 |
| GE (HADE) | 0.0269 | 0.0355 | 0.0444 |
| D (ADE) | 0.02 | 0.017 | 0.051 |
| GE (ADE) | 0.080665 | 0.072521 | 0.098227 |

**Table 6.** The root-mean-square errors (RMSE) of fADE and HADE at three iron concentrations.

| Concentration (Iron, mmol/L) | 0 | 0.0125 | 0.025 |
|---|---|---|---|
| RMSE (fADE) | 0.0257 | 0.0225 | 0.0363 |
| $\alpha$ (fADE) | 0.72 | 0.6 | 0.4 |
| RMSE (HADE) | 0.00665 | 0.0081 | 0.0124 |
| $\alpha$ (HADE) | 0.4125 | 0.6480 | 0.7647 |

## 4.1. The Effect of HA on Arsenic Transport

First, we quantified the migration of As(V) carried by the DOM-Fe ($Fe^{3+}$) composite colloids at different HA concentrations of 0, 5 mg/L, and 10 mg/L, respectively. For comparison purposes, the classical ADE model, and the HADE model were fitted with the same flow velocity. The experimental and simulation results are shown in Figure 2, while the simulation related parameters and errors are shown in Table 2.

The previous studies showed that DOM regulated arsenic transport, which can either facilitate or inhibit the migration [34,39]. The experimental results (Figure 2a) showed that under a certain ferric ion concentration and neutral pH conditions, the arsenic transport behavior is enhanced with the increase of HA concentration. This may be because increasing HA concentration makes reduces the size of colloidal particles, and promote the competition with As(V) for adsorption sites [40], and meanwhile, the number of ferrihydrite colloid loaded smaller chain HA than granular HA [41], thereby promoting arsenic migration [33]. The modeling results in Figure 2(b1–d2) show that the value of $\alpha$ (less than 1) increases with increasing HA concentration, meaning that the sub-diffusion behavior of arsenic is reduced with increasing HA concentration. The result graphs also show the difference between the starting point of the models and the experiment, which may be due to the residual solutions caused by experimental error and differences between the actual migration speed and the set migration speed in the actual migration process. In addition, from comparing the global errors and graphical results, the HADE model has advantages over the ADE model in simulating the arsenate migration, especially the early starting period, which has a certain delay effect; both the HADE model and the classical model can simulate the overall trend of the observed As(V) BTCs.

## 4.2. The Effect of pH on Arsenic Transport

Second, we discuss the migration of As(V) carried by the DOM-Fe ($Fe^{3+}$) composite colloids at different pH values of 4.5, 6.0, and 7.5.

As can be seen from Figure 3, the breakthrough curve shows a linear rise at pH 4.5. Then, as pH increases, the turbulence ratio and mobility of arsenic increase, promoting arsenic transport. It was

verified that co-deposition with ferric humate, As(V) transport, is facilitated under the neutral and alkaline pH conditions, while the transport is inhibited under acid condition [40]. Compared with the traditional model, the HADE model can better describe the reduction of deposition under neutral and alkaline conditions. While the ADE model did not give good simulation results for the deposition phenomena under acidic conditions, the HADE model showed excellent applicability. Meanwhile, with the increase in pH, the dispersion effect increases yet the sub-diffusion behavior is suppressed [33] due to the increasing trend in the value of $\alpha$ and the decrease in the D. From both, the simulation errors and the experimental graph results, the HADE model showed absolute superiority to the ADE model on the breakthrough curves which simulate the effect of pH on arsenic transport.

### 4.3. The Effect of Sodium on Arsenic Transport

Now, the migration of the DOM-Fe ($Fe^{3+}$) composite colloids carrying As(V) was checked under different sodium concentrations of 0.001 mol/L, 0.01 mol/L, and 0.1 mol/L.

Figure 4(b1,b2) indicates that when the sodium concentration is 0.001 mol/L, the experimental run showed significant noise in the measured arsenic concentrations, resulting in high uncertainty in model evaluation. Nevertheless, it was clear that as the sodium concentration increased, $\alpha$ value increased, the intensity of sub-diffusion decreased, and the arsenic migration velocity was reduced, meaning that the migration capacity of As(V) was suppressed [33]. The comparison results of the models in Figure 4b–d showed that both the HADE model and the ADE model can simulate the overall trend and tail of the experiment, but the advantages of the HADE model are still dominant, especially in the front delay part. Although the HADE model could get a very good fit when the noisy concentration was 0.001 mol/L, the global error value of the HADE was still smaller than the ADE model.

### 4.4. The Effect of Ferric Nitrate on Arsenic Transport

Finally, we apply the models to fit the observed migration of As(V) carried by the DOM-Fe ($Fe^{3+}$) composite colloids under different ferric nitrate concentrations of 0, 0.0125 mmol/L, and 0.025 mmol/L, and the simulation results are shown in Figure 5.

Hu et al. [42] indicated that the rate of As(V) morphological change is related to the ferrite transformation rate and that the overall reaction is revealed by the presence of HA adsorbed. Increasing the cationic metal content in the DOM by increasing the concentration of Fe ions in the DOM will increase the degree of complexation and the sub-diffusion intensity, while the interaction with HA will enhance competition with the arsenic adsorption site, and ultimately promote arsenic migration. Figure 5 showed that the iron promotes the migration of As(V), and the migration velocity increases with an increasing ferric ion concentration (the actual migration rate in the experiment is different from the set peristaltic pump speed), and the anomalous behavior is enhanced [33]. Meanwhile, the comparison results of the model simulator showed the superiority and accuracy of the HADE model, the advantage of simulating early delay behavior is extremely obvious. In simulating the migration of arsenic under different iron concentrations, the HADE model is not only superior to capture the observed As(V) BTC by the ADE model, but also to the simulations of the other three variables.

### 4.5. The Comparison between Fractional Model and HADE

The experimental data has been evaluated by using a time-fractional model (fADE) [33]. In this section, we would like to compare the proposed model (HADE) with the fADE model to discover its priority. For better comparison, the experiments of the effect of $Fe^{3+}$ on arsenic transport was selected, in which both models have shown their best fitting results. Figure 6 shows the comparison results of the two models and Table 6 gives the root-mean-square errors of the two models.

From a physical point of view, both HADE and fADE models are designed to characterize the influence of complex structures on solute transport. The results shown in Figure 6 and Table 6 clearly suggest that both HADE and fADE models can fit the experimental data. But HADE model performs better in describing the migration of arsenic influenced by iron, and its RMSEs are much smaller than

fADE. In addition, the expression of HADE (based on scale transfer) is simpler than the fADE model, because the fADE model involves convolution integral. It is worthy to mention that the HADE model is easy-to-implement in real-world applications. Because the Hausdorff fractal derivative is a local operator while the fractional derivative is a global operator.

## 5. Conclusions

This study establishes a HADE model to describe the migration of arsenic (As(V)) humic acid complex colloids in saturated quartz sand affected by HA, pH, the ion (sodium) strength, and the iron ion ($Fe^{3+}$) concentration. The HADE model uses the time Hausdorff fractal derivative with a varying fractal time index $0 < \alpha < 1$ to replace the integer-order derivative in the classical ADE when characterizing the anomalous transport behavior in arsenic migration. Under our experimental settings, pH, HA concentrations, and ferric ion concentration relative to sodium ion concentration have a significant effect on arsenic transport, which are consistent with the results in our previous work [33]. The main findings of this paper are listed below.

1. The physical mechanism in As(V) transport in this heterogeneous porous medium can be explained by the time HADE model. From the results above, the intensity of the sub-diffusion behavior of arsenic migration was negatively correlated with the increase of pH, HA concentration, and sodium ion concentration, and positively correlated with the increase of ferric ion concentration, which can be well reflected by the change of $\alpha$.
2. The HADE model has more obvious advantages than the ADE model in simulating arsenic migration. Compared with the fADE model (used in [33]), the advantages of the HADE model include an accurate description (especially at the early stage), simple expression, and easy-to-implement.
3. By comparing the mean global errors in the simulation results of four variables ($Fe^{3+}$, pH, HA, and $Na^{+}$) in affecting arsenic migration, the best sorting based on the HADE fitting effect is $Fe^{3+} > pH > HA > Na^{+}$.

Our future work will focus on employing the Hausdorff fractal models to predict transport behaviors of other pollutants with or without chemical reactions and compare them with other extended models. Our ultimate goal is to establish a new mathematical scheme to simulate solute transport, especially heavy metal transport.

**Author Contributions:** Conceptualization, methodology, H.S. and S.L.; data collection: S.L., J.S., and X.H.; writing—original draft preparation, X.H.; writing—review and modification, H.S., Y.Z., and K.S.; funding acquisition, H.S. and X.H.; supervision, H.S. All authors have read and agreed to the published version of the manuscript.

**Funding:** This research was funded by the Natural Science Foundation of Jiangsu Province, P.R. China, grant number BK20190024, the Fundamental Research Funds for the Central Universities, grant number 2019B65114, Postgraduate Research & Practice Innovation Program of Jiangsu Province, grant number SJKY19_0420, and the National Natural Science Foundation of China, grant numbers 11972148 and 41931292.

**Acknowledgments:** We thank two anonymous reviewers for their valuable comments.

**Conflicts of Interest:** The authors declare no conflict of interest.

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
