# Peer review of "Hausdorff Fractal Derivative Model to Characterize Transport of Inorganic Arsenic in Porous Media"

_water, doi:10.3390/w12092353_

Round 1

Reviewer 1 Report

I think that improving the grammatical structure would significantly improve the quality of the paper and get the point across easily.   L150: Typo? “to” missing? L188: “technology”? Maybe rephrasing might improve clarity of thought here.

Author Response

We would like to thank the two reviewers the constructive suggestions and comments that significantly improved the presentation of this work. We considered every comment and revised the manuscript correspondingly. The questions raised by the reviewers were addressed in the revised manuscript (highlighted by red color). Our itemized response is also presented below.

Reviewer #1 (Written Evaluation (to Author) (Required)):

Comment1: I think that improving the grammatical structure would significantly improve the quality of the paper and get the point across easily.  

Response: Thank you, we improved the grammatical structure of the paper.

Comment 2: L150: Typo? “to” missing?

Response: Thank you, we added the missing “to”. (L140)

Comment 3: L188: “technology”? Maybe rephrasing might improve clarity of thought here.

Response: Thank you, we changed the “technology” with “method”. (L176)

Reviewer 2 Report

First, I would like to congratulate the authors on making considerable improvements, particularly to the results section of this manuscript, in a relatively short amount of time. The updated comparisons between the Hausdorff fractal derivative model and the classic ADE (α = 1) make a very clear case for the benefit of applying this model over the standard technique. The inclusion of a general error (GE) comparison term to assess the goodness of fit for each model allows for robust comparison of the two different modelling approaches. Minor improvements have also been made to the introduction, materials, and methods sections of the paper.

Despite these substantial improvements, I do not feel comfortable recommending this manuscript for publication due to the direct replication of experimental results from Yao et al., Chemosphere. 2019, 240 (2020), 124987. The transport data presented in this manuscript (water-850504) uses identical experimental conditions, tests the exact same factors, and presents the exact As(V) breakthrough data to the point that when overlaid with the corresponding figure from Yao et al., the results perfectly match. Please see Figure 1 (available in the PDF version of these comments) embedded below for an example. Note that this can be done for every graph presented in water-850504.

(See PDF version of author comments for figure)

Figure 1. Overlaid images from water-850504 Figure 4 (semi-transparent) and Yao et al. Chemosphere. 2019, 240 (2020), 124987 Figure 6 (solid). The brighter data points show those that overlap between the datasets presented in the papers. Note that for every data point presented in Yao et al., there is a corresponding, exact, matching datapoint for the figure from water-850504.

Furthermore, the discussion and conclusions of water-850504 still do not reference the Yao et al. paper (which is only referenced in the methods section to justify parameter selection). Since the analysis in water-850504 appears to be a re-evaluation of existing experimental data, I do not find it acceptable to present the results as new experiments, and conclusions should be discussed expressly in relation to the Yao et al. paper (the source of the data).

I would like to communicate to the authors that I think the modelling effort described in this work is interesting and valuable, and that the robust mathematical comparison between the HADE and the classic ADE is highly convincing. However, the ethical concerns I presented over the source of the data for this manuscript prohibit me from making a recommendation for publication. If the editor agrees on re-submission for a 3rd evaluation, for me to recommend publication, the following criteria will have to be met:

  1. The methods must directly state that an existing dataset is being re-evaluated in this work
  2. The results must discuss the performance of the HADE model in relation to the fADE model (which has already been evaluated on the existing dataset)
  3. The conclusions must focus on the performance of the HADE model compared to the ADE and fADE. Any mention of the influence of parameters ([HA], IS, pH, [FeNO3]) must compare to the Yao et al. paper (which has already established the influence of these parameters using the same dataset)

Author Response

This manuscript is a resubmission of an earlier submission. The following is a list of the peer review reports and author responses from that submission.

Round 1

Reviewer 1 Report

In this paper the authors added Hausdorff fractal model to the ADE to improve modeling and prediction accuracy compared to experiments. The paper is interesting, but needs work done to make it clear and get the point across.   One thing I would emphasize is please refrain from using too many acronyms. The write-up gets confusing at best. There are also a lot of typos like missing spaces which need to be corrected.   L44-47: Better to write “arsenic” instead of As L47: Typo “As under” L92: What is “DOM”? Is it dissolved organic matter? L102: Is it 0.01M or 1M NaNO3 solution? L114: Please add an illustration of the experimental setup to clarify the setup. L115: I would suggest to create a graphic of the 4 runs with the parameters that have been varied. That would make the it easier to compare and understand. Figs 1,2,3,4: Please tabulate the fitting parameters (alpha and D) for the three concentrations Fig 1: Please comment on the difference between the starting point of the model and experiment. Why does the ADE/HADE model starts increasing at ~7-10 mins while the experimental results start increasing at ~25 mins? Fig 3b: Please comment on the sudden jump in arsenic concentration at ~350 mins L243: “wakened” typo? L264: “… as the concentration” typo?

Reviewer 2 Report

            The manuscript details the preparation of arsenate/DOM/ferric ion solutions and subsequent injection into quartz sand packed column. The experimental arsenate breakthrough data from various trials evaluating the effect of factors including pH, humic acid concentration ([HA]), ionic strength, and iron concentration are presented. A Hausdorff fractal derivative time variable (tα) replaces the classic time variable (t) in an advection-dispersion equation to produce a new “HADE” model. This new HADE model is used to fit the experimental BTCs and is compared directly to the ADE models without the fractal derivative time variable. Based on the values of the fractal index, α, and the BTC observations, some conclusions are drawn regarding the effect of pH, [HA], ionic strength, and [Fe] on arsenate mobility in porous media.

            While the HADE model is mathematically sound and provides reasonable fit for the arsenate BTC data, meaningful, quantitative comparison between the HADE and ADE models is absent from the publication. Furthermore, it may be unsurprising that the ADE is insufficient for adequately representing arsenate/DOM/Fe colloid transport since it does not incorporate any reactive transport components. The Hausdorff fractal component of the model accounts for subdiffusion by use of its temporal fractional derivative, for which lower α values represent increased solute retention. There exist numerous modified ADEs which consider variations on reactive solute transport and it would be more interesting to evaluate the accuracy and simplicity of this new HADE in comparison to these established models. Considering this evaluation, I’m afraid I cannot recommend publication of the manuscript in its current form. Substantial improvements, examples of which are provided below, should to be made to all areas of the manuscript to render it suitable for publication.

Broad Comments:

Strengths:

  • The application of a temporal Hausdorff fractal derivative to the ADE is an interesting and fairly simple improvement to make to the classic advection dispersion equation

Weaknesses:

  • The introduction of the paper does not adequately prepare the reader to understand the importance of the HADE model compared to the state of the art or the fundamentals of arsenate/DOM/Fe colloid transport
  • The materials and methods section is poorly written and does not provide adequate description of the experiments. The presentation of transport solution formulations is not clear and some information is missing.
  • The figures are poorly constructed and do not allow easy visual comparison of the breakthrough curves from the various experiments.
  • The HADE and ADE models are not adequately quantitatively compared to allow a definitive conclusion over which model is superior.
  • The understanding of factors influencing arsenate transport in the presence of dissolved organic matter and ferric iron has already been elucidated in previous articles (for example, reference 31 of this paper, Chemosphere. 2019, 240 (2020), 124987). The conclusions of this manuscript should only address the fit and interpretation of the new HADE model.

Specific Comments:

Introduction:

  • Numerous grammatical and spelling errors are made throughout the introduction and throughout the paper. For example:
    • Line 31: ‘chemical’ should be replaced with ‘element’
    • Line 34: ‘serious’ should be replaced with ‘series’
    • Line 39: ‘To exploring’ should be ‘to explore’
  • The entire paper should be re-evaluated for grammar and spelling
  • Lines 39-47: background on the formation of iron oxy-hydroxide/humic acid colloids in solution is not sufficiently described here. The spontaneous precipitation of these colloids from ferric iron in solution is critical to understanding arsenate transport in this system.
  • Lines 62-79: this section should aim to familiarize readers with the Hausdorff fractal derivative. It would be appropriate to move equations (1) and (2) from section 3 into the introduction. Reference findings from Chen et al (reference 28) and describe why only substitution of the temporal variable is necessary.
  • Discuss the recent findings of Yao et al. (reference 31) using the fADE model and describe how the HADE model is different or improves on this modelling effort.

Methods:

  • Add a materials section at the beginning to describe the source and properties of the chemicals and reagents used.
  • Titles of the various sections should be improved
    • e. Granular Porous Media Pretreatment → Porous Media Preparation and Column Packing; The Experimental Runs → Arsenate Transport Experiments
  • Lines 97-102: preparation of stock solutions should be provided with appropriate detail
    • e. Third, 1 mmol/L iron nitrate stock solution was prepared by adding iron nitrate (nonahydrate?) to (Millipore deionized water)
  • Line 110: the column is not made of quartz sand, as described. What material is the column made of?
  • Lines 111-114: specify the dry-packing method. The description of PV measurement is unclear. Was the PV simply estimated based on the mass of the packed sand?
  • Lines 116-133: the text explanation of the various solution formulations is long and confusing. This information could be concisely summarized in a table
  • Line 120: how was the pH adjusted? NaNO3 should not be sufficient for acidifying the solution, yet no acids are mentioned.
  • There is no mention of the methodology for assessing arsenic concentration in the BTCs

'Hausdorff Fractal Derivate Model'

  • The title uses the wrong word “Derivate” instead of “Derivative”
  • Equation (2) contains an error in the limit for the temporal variable, the alpha index should not be present
  • For Equations (3) and (4), it is recommended to use C for concentration and omit the superfluous term (currently C) for hydraulic conductivity
  • Lines 148-149: provide units for the described variables
    • e. u is the average fluid velocity [m/s], etc.
  • Lines 154-155: make it clear that the analytical solution to the HADE is simply the analytical solution to the ADE with the time variable substituted for tα.
  • Add a description of the dispersion coefficient in the HADE which has a time dimension that should be modified by α. Explain that it will differ from D in the classic ADE.
  • For equation (7) the boundary conditions are awkwardly mathematically stated.
    • Correct format should be lim x→∞ φ(x,t)=0

Results:

  • All figures are poorly constructed and comparison between the BTCs is impossible the way the figures are presented.
    • All BTCs should be plotted on the same graph for comparison
    • Modelled results can be added in additional panels to avoid cluttering the BTC graphs
    • RMSE or sum of square error information for each model fit should be provided to evaluate model goodness of fit for the ADE and HADE models
  • Tabulated results of all experiments should be provided including alpha, D, and RMSE values for each model. Dimensions should be specified for D values.
  • Results in each figure are very unconvincing for the analysis and conclusions presented. For example, lines 171-173 indicate that As transport is enhanced at higher HA concentrations, but panel a of Figure 1 shows that As transport in the 0 mg/L HA condition is nearly identical to As transport in the 5 mg/L and 10 mg/L conditions.
  • The Classic ADE is unsuitable for describing reactive transport, but various modified versions exist which account for kinetic attachment/detachment behaviour of colloids. It would be interesting to compare the new HADE model not only with the classic ADE, but also with modified advection-dispersion-reaction equations to evaluate the value of this new model compared to the state-of-the-art.
    • The HADE should also be compared with the fADE from reference 31 (Yao et al, Chemosphere) which is an almost identical model. Is the HADE an improvement on this model in any way besides being less computationally intensive?

Conclusions:

  • The conclusions regarding the HADE model are suitable, but the conclusions regarding the influence of the parameters pH, [HA], [Fe], and ionic strength have already been established in previous articles, rendering their conclusion here redundant. This manuscript should focus solely on the evaluation of the new HADE model and only make conclusions based on new findings.